# Neonatal Microcephaly and Central Nervous System Abnormalities During the Zika Outbreak in Rio de Janeiro

**DOI:** 10.3390/v17020208

**Published:** 2025-01-31

**Authors:** Marlos Melo Martins, Roberto de Andrade Medronho, Carlos Eduardo Raymundo, Arnaldo Prata-Barbosa, Antonio José Ledo Alves da Cunha

**Affiliations:** 1Division of Pediatric Neurology, Martagão Gesteira Institute of Childcare and Pediatrics, Federal University of Rio de Janeiro (UFRJ), Rio de Janeiro 21941-912, Brazil; marlosmelo@ippmg.ufrj.br; 2Department of Epidemiology and Public Health, School of Medicine, Federal University of Rio de Janeiro (UFRJ), Rio de Janeiro 21941-592, Brazil; medronho@iesc.ufrj.br (R.d.A.M.); caducer@gmail.com (C.E.R.); 3Department of Pediatrics, D’Or Institute for Research and Education (IDOR), Rio de Janeiro 2281-100, Brazil; 4Department of Pediatrics, School of Medicine, Federal University of Rio de Janeiro (UFRJ), Rio de Janeiro 21044-020, Brazil; acunha@hucff.ufrj.br

**Keywords:** Zika virus, microcephaly, CNS congenital malformations

## Abstract

This retrospective cohort study analyzed 7870 pregnant women, including 2269 with confirmed Zika virus (ZIKV) infection and 5601 without Zika infection, along with their fetuses and newborns. Data were sourced from multiple databases in the state of Rio de Janeiro, Brazil. A propensity score model was employed to control confounding factors and stratify outcomes by pregnancy trimester. Among ZIKV+ pregnant women, 49 cases of congenital microcephaly or congenital nervous system (CNS) abnormalities were identified (2.16%, or 193.9 cases in 10,000 live births), whereas 44 cases were identified among ZIKV− women (0.78%, or 71.4 cases in 10,000 live births). Multivariable analysis yielded an odds ratio of 2.46 (95% CI 1.30–4.64) overall, with 4.29 (95% CI 1.93–9.53) in the first trimester, 5.29 (95% CI 1.08–25.95) in the second trimester, and 0.68 (95% CI 0.21–2.14) in the third trimester. The most frequent findings among ZIKV+ cases included intracranial calcifications, ventriculomegaly, posterior fossa malformations, reduced brain volume, corpus callosum malformations, cortex dysplasia, lissencephaly, and pachygyria. Ophthalmologic abnormalities were detected in 55.5% of cases, and brainstem auditory evoked potential anomalies were reported in 33.3%. ZIKV infection can result in structural or functional anomalies. Given the absence of specific treatment for congenital Zika syndrome (CZS), clinical care should prioritize monitoring and managing neurological, motor, auditory, visual, and orthopedic disorders in all children with in utero ZIKV exposure, especially during the first and second trimesters of pregnancy.

## 1. Introduction

Over the past ten years, the Zika virus (ZIKV) has transitioned from being associated with mild infections to becoming one of the most extensively studied viruses globally. Between 2015 and 2016, ZIKV caused a significant outbreak in the Americas, particularly in Brazil. During this period, initial reports emerged of pregnant women with confirmed or suspected ZIKV infection giving birth to fetuses and newborns with severe congenital malformations, most notably microcephaly and central nervous system (CNS) abnormalities. These findings strongly suggested an association with the virus, which has since been supported by accumulating evidence [1,2,3].

As of January 2018, over 3700 cases of congenital Zika syndrome (CZS) had been reported in the Americas [4]. Local transmission of the Zika virus (ZIKV) has been documented in 87 countries and territories worldwide, spanning tropical and subtropical regions. Since the end of 2016, ZIKV transmission has significantly declined, with reported cases decreasing from more than 500,000 in 2016 to fewer than 30,000 in 2018 [5].

ZIKV outbreaks continue to emerge in various regions of the world, including India and Southeast Asia, where large populations of women and their infants remain vulnerable to ZIKV infection [6]. This ongoing threat represents a significant public health concern, as ZIKV transmission can occur during non-epidemic periods, with most cases in pregnant women being asymptomatic but still posing risks to their fetuses and newborns [6,7]. This study aimed to deepen our understanding of the ZIKV outbreak in the state of Rio de Janeiro and its impact on pregnant women and their infants.

## 2. Materials and Methods

### 2.1. Study Design and Setting

This retrospective cohort study utilized data from public health databases collected between February 2015 and December 2018, focusing on pregnant women, their fetuses, and newborns during the ZIKV outbreak in Rio de Janeiro, Brazil. Five databases were used: GAL, FORMSUS, RESP, SINASC, and SIM. GAL is an online laboratory system that provides real-time reverse transcriptase polymerase chain reaction (RT-PCR) test results for ZIKV, along with patient characteristics. FORMSUS is an online platform used to collect, store, and generate health data reports, including information on pregnant women. RESP is an online notification system used to report suspected cases of microcephaly or CNS congenital abnormalities during the ZIKV outbreak in Brazil. SINASC records all registered newborns in Brazil, while SIM documents deaths across the country. All databases were provided by the Health Secretary of Rio de Janeiro, and the study protocol received ethical approval in accordance with the Code of Ethics of the World Medical Association (Declaration of Helsinki) (Institutional Review Board number 87402618.3.0000.5275).

We initially used the GAL database to identify two cohorts of pregnant women. Eligibility for the confirmed ZIKV infection cohort required at least one positive RT-PCR result for ZIKV in blood or urine samples collected during pregnancy. For the non-ZIKV infection cohort, we included pregnant women with consistently negative tests, with samples collected within five days of symptom onset for blood and within fourteen days for urine. The RESP database was then reviewed on a case-by-case basis to identify congenital CNS abnormalities and microcephaly cases. Cases initially reported but not meeting the criteria for congenital microcephaly, as defined by the Intergrowth21st method, were excluded. This step was necessary due to the varied definitions used during the early stages of the ZIKV outbreak in Brazil. Finally, a probabilistic linkage—a statistical method to integrate data from multiple sources—was performed to consolidate information across the five databases.

From the final consolidated database, we extracted maternal covariates, including age, race, marital status, education level, region of residence, the presence of twin pregnancies, type of labor, number of medical appointments during pregnancy, and the need for hospitalization, as well as signs and symptoms reported during pregnancy. For the newborns, in addition to identifying the presence of microcephaly and CNS abnormalities, we collected data on gender, birth length, weight, cephalic perimeter, death, prematurity, results of serologic tests other than ZIKV, RT-PCR results obtained from the newborns, type of CNS abnormalities (if present), physical examination findings, and the presence of ocular and auditory anomalies.

### 2.2. Statistical Analysis

Descriptive statistics were generated for all baseline data. Prevalence rates were calculated by dividing the number of cases of microcephaly or CNS congenital abnormalities by the total number of pregnant women in each group. Outcomes are reported both overall and stratified by the trimester of symptom onset. Continuous variables are summarized as medians with interquartile ranges (IQRs), while categorical variables are presented as proportions (%).

Categorical variables were compared between women and infants with positive and negative ZIKV RT-PCR results using the Chi-square test, Mann–Whitney test, or Fisher’s exact test, as appropriate. A *p*-value of less than 0.05 was considered statistically significant. Propensity-adjusted analyses were conducted to control for measured confounding variables. The propensity score method was employed, where the outcome variable was regressed on an indicator variable representing exposure status and the estimated propensity score. Given the dichotomous nature of the outcome, a regression model was used, with the exposure effect in the logistic model expressed as an adjusted odds ratio (OR). All measured baseline covariates were included in the propensity score model. Odds ratios and their 95% confidence intervals (CIs) were calculated, with a *p*-value of less than 0.05 considered statistically significant.

## 3. Results

### 3.1. Study Population

From February 2015 to December 2018, we identified 2269 pregnant women with at least one positive result for ZIKV infection on RT-PCR in blood, urine, or both (ZIKV+), and 5601 pregnant women with only negative ZIKV infection tests (ZIKV−). Among ZIKV+ women, 758 had positive RT-PCR results in serum specimens, 88 in urine, 270 in serum and urine specimens, and 24 in more than one serum specimen. In those women with positive and negative results, 728 were only positive in serum, and 401 were only in urine. The most common signs and symptoms among ZIKV+ pregnant women were rash, pruritus, headache, arthralgia, myalgia, and fever. Despite that, ZIKV− pregnant women showed a predominance of headaches, arthralgia, myalgia, fever, edema, and coryza (Table 1).

Of the ZIKV+ pregnant women, 49 cases of congenital microcephaly or CNS abnormalities were identified (2.16%). In the ZIKV− group, 44 cases were identified (0.78%). The prevalence among live births was 193.9 in 10,000 exposed to ZIKV (five fetal losses) and 71.4 in 10,000 not exposed to ZIKV (four fetal losses). The prevalence varied among trimesters when the ZIKV infection occurred: 5.5% in the first trimester, 1.5% in the second, and 0.79% in the third (Table 2).

The baseline characteristics were quite similar between the women with and without ZIKV infection. Most women were aged 21–31, single, lived in the metropolitan area, and had a high school education. Cesarean section deliveries were more frequent in both groups, and they had at least six medical appointments during pregnancy (Table 3).

### 3.2. Multivariable Analysis

The odds ratio of congenital microcephaly or CNS abnormalities caused by exposure to ZIKV during pregnancy was 2.46 (95% CI 1.30–4.64). Pregnant women were also stratified in the three different trimesters when the ZIKV infection occurred or was suspected: 4.29 (95% CI 1.93–9.53) in the first trimester, 5.29 (95% CI 1.08–25.95) in the second trimester, and 0.68 (95% CI 0.21–2.14) in the third trimester (Table 4).

### 3.3. Adverse Prenatal and Early Postnatal Infant Outcomes

Among the 93 cases of congenital microcephaly or CNS abnormalities among ZIKV+ and ZIKV− pregnant women, the median maternal age was quite the same in both groups. We did not see statistical differences in maternal race, residence, sex, weight, and length of the newborns. The median and IQR of head circumference were also very similar, as were prematurity, twinning, and fetal loss (Table 5). Maternal symptoms, such as rash and pruritus, were more prevalent in the ZIKV+ group (Table 6).

Thirty-five of the 49 cases of ZIKV+ pregnant women (71.4%) and 20 of the 44 cases of ZIKV− women (45.5%) underwent at least one imaging exam: fetal or cranial ultrasound, cranial computed tomography, or magnetic resonance imaging (MRI). The most common findings among ZIKV+ cases were intracranial calcifications, ventriculomegaly, posterior fossa malformations, reduced brain volume, corpus callosum malformations, cortex dysplasia, lissencephaly, and pachygyria. ZIKV− cases had very similar findings, although ventriculomegaly was more frequent in this group. Arthrogryposis was the most common physical examination finding. Ophthalmologic exams were performed in nine ZIKV+ newborns (55.5% abnormal) and ten ZIKV− newborns (30% abnormal). Brainstem auditory evoked potentials (BAEPs) were reported in nine ZIKV+ cases (33.3% abnormal) and in six ZIKV− cases (16.6% abnormal) (Table 7).

## 4. Discussion

Our main evidence is that pregnant women infected with ZIKV are at increased risk of having fetuses or newborns with microcephaly or CNS congenital abnormalities when compared to pregnant women with no ZIKV infection, with six to seven times the risk when the infection occurs in the first and second trimesters compared to those in the third trimester.

The signs and symptoms of ZIKV infection among pregnant women in our cohort were consistent with findings from other studies. Garcell et al. [8], analyzing 1541 patients with clinical suspicion of arbovirosis, and Tozetto–Mendonza et al. [9], who studied 94 patients with acute ZIKV infection confirmed by RT-PCR, reported that rash, pruritus, arthralgia, headache, and myalgia were the most common symptoms. Similarly, published cohorts of pregnant women with confirmed or probable ZIKV infection also highlighted the predominance of these symptoms, along with fever, conjunctivitis, and retro-ocular pain [10,11,12,13,14,15,16,17]. Braga et al. [18] developed a predictive score model to differentiate symptoms of acute ZIKV infection in regions with co-circulating arboviruses. Their model achieved 86.6% sensitivity and 78.3% specificity for rash associated with pruritus or conjunctival hyperemia in the absence of fever, petechiae, or anorexia. Moreover, ZIKV infection can be asymptomatic in 50–73% of cases in the general population [7,19], and approximately 69% of pregnant women may remain asymptomatic [12]. Therefore, CZS can occur even in those asymptomatic pregnant women. Meneses et al. [20] reported that 24% of children with CZS, with maternal infection confirmed by RT-PCR, had no reports of symptoms during pregnancy. Vianna et al. [21] also report that 17% of children referred for CZS investigation had no history of maternal symptoms during pregnancy.

The differential etiological diagnosis of microcephaly or CNS congenital abnormalities extends beyond ZIKV and must include other infections such as dengue, chikungunya, toxoplasmosis, cytomegalovirus (CMV), syphilis, human immunodeficiency virus (HIV), parvovirus B19, measles, rubella, chickenpox, herpes virus, Epstein–Barr virus, and enterovirus [22]. Although less common, coinfections should always be considered. We observed a predominance of arthralgia, headache, coryza, edema, fever, and myalgia among pregnant women without ZIKV infection. Dengue typically presents a more exuberant fever associated with rash, headache, and myalgia, while chikungunya is characterized by high fever with rash, polyarthralgia/polyarthritis, myalgia, and edema [23]. Vertical transmission of the dengue virus can occur in pregnant women with viremia at the time of delivery, and it can also be found in breast milk [24,25]. Chikungunya virus can also cause neonatal infection in up to 50% of pregnant women with viremia in the peripartum period [26], ranging from mild to severe cases and lethality in around 2.8% [26,27]. While an association between dengue virus and congenital microcephaly or CNS abnormalities has not been described, postnatal microcephaly linked to chikungunya virus infection has been reported. In a series of 25 brain MRIs of congenital chikungunya virus infections, intraparenchymal hemorrhages and white matter abnormalities were observed in 14 cases [28], though congenital microcephaly was absent. However, postnatal microcephaly is described [29]. Infectious mononucleosis is another possibility, with a higher fetal risk of death and congenital malformations, especially those associated with toxoplasmosis and CMV. Parvovirus B19, rubella, measles, and enteroviruses (mainly coxsackie A and B) also increase the risk of fetal mortality and congenital malformations [22].

The 2.16% prevalence of congenital microcephaly or CNS abnormalities observed in our cohort is lower than that reported in other studies [30,31]. Several factors may explain this difference: (1) only pregnant women with a positive ZIKV RT-PCR result were included; (2) the majority of these women (83.7%) were symptomatic, resulting in a smaller proportion of asymptomatic women, who may also carry a risk of congenital abnormalities in their fetuses; (3) the use of retrospective data; and (4) the absence of follow-up data, which could underestimate the prevalence due to the possibility of postnatal manifestation of signs and symptoms [32,33].

The chance of congenital microcephaly or CNS abnormalities following ZIKV exposure is inversely proportional to the trimester in which the infection occurs [10,34,35,36,37,38]. Like other congenital infections, ZIKV infection during the first trimester poses the highest risk for congenital CNS injuries. This elevated risk is likely due to the greater extent of cell loss in both placental and fetal tissues. Histopathological findings from fatal cases in fetuses with microcephaly and a maternal history of ZIKV infection during the first trimester include villous edema, an increase in the number of Hofbauer cells, and the presence of antigens in the chorionic villi and necrotic fetal nerve cells or during the degeneration process, as well as in glial cells [39,40]. The presence of neuronal necrosis indicates ongoing cell injury, extending from the period of maternal infection to subsequent stages of brain development [39]. The loss of nerve cells in the early stages of CNS development can result in less brain volume and impair the formation of cortical gyri [41,42]. A recently published Brazilian cohort demonstrated a higher chance of developmental abnormalities between three and five months of age in pregnant women with ZIKV infection during the first trimester [32]. Interestingly, the lower odds ratio found in the first trimester compared to the second may reflect a higher prevalence of adverse outcomes in negative ZIKV pregnant women, potentially caused by other infections, environmental factors, or even abnormalities of genetic origin.

Laboratory confirmation of CZS is more challenging, particularly because the duration and pattern of ZIKV viremia and viruria—whether constant or intermittent—are not yet well understood, especially in those infected in the first trimester of pregnancy. This uncertainty may explain the significant variation in ZIKV RT-PCR’s positivity in different body fluids (blood, urine, and liquor amnii) across cohorts, ranging from no positive tests to 65% positivity [10,35,38,43,44,45]. Serological testing of newborns for ZIKV in the postnatal period, confirmed by the plaque reduction neutralization test (PRNT), demonstrates a sensitivity that varies from 7.1 to 90.5% [46,47,48] and can be influenced by the time of its realization, with a marked decline in sensibility after the first month of life [46]. Additionally, the fetal immune response to ZIKV may be like other congenital infections, such as rubella and CMV, which are characterized by a stronger cellular immune response and a comparatively humoral immune response [45].

The predominant findings of intracranial calcifications, ventriculomegaly, reduced brain volume, and cortical defects (lissencephaly and pachygyria) observed in our study are consistent with descriptions by other authors [49,50]. Intracranial calcifications associated with CZS are typically located in the subcortical regions. Recent reports show other potential sites such as infratentorial, base nuclei, periventricular region, and the cortex itself [51]. Ventriculomegaly was a frequent finding among our CZS cases, though it was even more prevalent among those born from women without laboratory evidence of ZIKV infection. It is noteworthy that only about 5% of congenital ventriculomegaly cases are attributed to congenital infections such as CMV, toxoplasmosis, and ZIKV [52]; therefore, the higher frequency observed in our study may reflect a result of other non-infectious etiologies.

The most common ophthalmological findings in CZS include pigmentary abnormalities and chorioretinal atrophy, which are similar to those observed in congenital toxoplasmosis [53]. Approximately 34 to 55% of children with CZS and microcephaly have at least one ophthalmic abnormality [54]. Despite the limited number of ophthalmic exams, 55.5% revealed abnormalities, including optic nerve hypoplasia, incomplete vascularization, and pigmentary abnormalities. Although not the most frequent finding, optic nerve hypoplasia has been reported by some authors [54,55]. In cases negative for ZIKV, alternative etiologies should be explored, such as other congenital infections (e.g., CMV and toxoplasmosis) or genetic and metabolic diseases [56,57]. Incomplete vascularization was described in a premature newborn since complete retinal vascularization occurs only at around 40 to 42 weeks of gestation [58].

Regarding auditory abnormalities, newborns exposed to ZIKV in our study exhibited a higher frequency of abnormalities compared to those not exposed (33.3% vs. 16.6%). Children exposed to ZIKV show wide variation in its frequency, whereas abnormalities in otoacoustic emissions vary from 0 to 75%, while brainstem auditory evoked potentials (BAEPs) vary from 0 to 29.2% [59]. Traditionally, other congenital infections (CMV and rubella) and genetic abnormalities are the leading causes of congenital hearing loss [60].

Arthrogryposis has also been described by other authors [61,62,63,64]. Approximately 80% of congenital arthrogryposis have a neurogenic origin, due to abnormalities in the formation, structure, or function of central and peripheral nervous systems [61]. Imaging and electroneuromyography studies have demonstrated this involvement of both nervous systems, with a volumetric reduction of the anterior medullary tracts in children with CZS, reducing fetal mobility and, consequently, causing deformities [63,65].

The strengths of our study include a large population sample, confirmation of ZIKV infection in pregnant women through RT-PCR, the inclusion of a control group, an extensive evaluation of potential covariates, the application of multivariate analysis, and a thorough review of all cases. However, the following limitations should be acknowledged: (1) use of retrospective data; (2) incomplete information for part of the sample, including serologic tests for dengue, chikungunya, CMV, toxoplasmosis, rubella, and herpes; (3) the lack of complementary exams among the identified cases; and (4) the predominance of symptomatic ZIKV-infected pregnant women in the cohort, which limits the generalizability of our findings to all pregnant women, including asymptomatic cases.

## 5. Conclusions

It is essential to highlight that ZIKV infection can lead to a spectrum of structural or functional anomalies. As there is no specific treatment for CZS, assistance should be focused on monitoring neurological, motor, auditory, visual, and orthopedic disorders in all children with intrauterine exposure to ZIKV, regardless of the identification of congenital anomalies in the prenatal period, especially to those exposed during the first and the second trimesters of pregnancy. ZIKV infections are still endemic in some countries globally, including Brazil, reinforcing the preventive measures and continuous monitoring of potential fetal anomalies.

## Figures and Tables

**Table 1 viruses-17-00208-t001:** Signs and symptoms in ZIKV+ and ZIKV− pregnant women.

Signs and Symptoms	ZIKV+ *n* (%)	ZIKV− *n* (%)	*p*-Value ^1^
Rash	1899 (83.7)	4477 (79.9)	0.0001
Pruritus	1391 (61.3)	2656 (47.4)	<0.001
Headache	636 (28.0)	1943 (34.7)	<0.001
Arthralgia	596 (26.3)	2196 (39.2)	<0.001
Myalgia	476 (20.1)	1670 (29.8)	<0.001
Fever	379 (16.7)	1862 (33.2)	<0.001
Retro-ocular pain	322 (14.2)	835 (14.9)	0.1164
Conjunctival hyperemia	195 (8.6)	414 (7.4)	0.2216
Edema	187 (8.2)	552 (9.6)	0.0053
Conjunctivitis	90 (4.0)	210 (3.7)	0.9846
Coryza	68 (3.6)	237 (5.3)	0.0041
Cough	79 (3.5)	236 (4.2)	0.0704
Diarrhea	127 (3.0)	337 (6.0)	0.2595
Lymphadenomegaly	49 (2.2)	107 (1.9)	0.718

^1^: Chi-square test.

**Table 2 viruses-17-00208-t002:** Prevalence of outcomes in ZIKV+ and ZIKV− pregnant women (total and in different trimesters).

	ZIKV+	ZIKV−	*p*-Value ^1^
Pregnant women	2269	5601	
Outcomes	49	44	<0.001
Prevalence	2.6%	0.78%	
**1st trimester**			
Pregnant women	508	1205	
Outcomes	28	19	<0.001
Prevalence	5.5%	1.6%	
**2nd trimester**			
Pregnant women	997	1941	
Outcomes	15	7	0.0015
Prevalence	1.5%	0.36%	
**3rd trimester**			
Pregnant women	764	2455	
Outcomes	6	18	1.0
Prevalence	0.79%	0.73%	

^1^ Chi-square test.

**Table 3 viruses-17-00208-t003:** Maternal characteristics in ZIKV+ and ZIKV− pregnant women.

	ZIKV+ *n* (%)	ZIKV− *n* (%)	*p*-Value ^1^
**Total**	2269	5601	
**Age** [median (IQR)]	26 (21–31)	26 (21–31)	0.233 *
**Ethnicity/race**			
White	945 (44.6)	2197 (42.8)	0.168
Others	1176 (55.4)	2937 (57.2)	
Missing	148	467	
**Place of residence**			
Not urban	352 (15.5)	765 (13.7)	<0.001
Urban	1917 (84.5)	4836 (86.3)	
**Marital status**			
Single	1019 (70.0)	2398 (72.2)	0.402
Married	411 (28.2)	872 (26.3)	
Widowed	3 (0.2)	4 (0.1)	
Divorced	23 (1.6)	46 (1.4)	
Missing	148	467	
**Education**			
Elementary school	289 (19.4)	627 (18.4)	0.331
High school	980 (65.6)	2219 (65.0)	
Higher school	224 (15.0)	566 (16.6)	
Missing	776	2189	
**Twin pregnancy**			
No	1501 (99.1)	3413 (98.7)	0.181
Yes	13 (0.9)	45 (1.3)	
Missing	755	2143	
**Type of labor**			
Natural	668 (44.1)	1596 (46.2)	0.179
Cesarean section	847 (55.9)	1862 (53.8)	
Missing	754	2143	
**Prenatal consultations**			
≥6	1295 (86.9)	2920 (85.8)	0.318
<6	196 (13.1)	484 (14.2)	
Missing	778	2197	
**Need for hospitalization**			
No	1686 (98)	3810 (97.6)	0.388
Yes	35 (2)	94 (2.4)	
Missing	548	1697	

Percentages for all categories were calculated with the exclusion of those with missing data from the denominator. The “missing” category was not included as a category when the *p*-value was estimated. IQR: interquartile range. ^1^: Chi-square test, except for the one labeled with an asterisk (*), which used Fisher’s exact test.

**Table 4 viruses-17-00208-t004:** Average effect of exposure to ZIKV during pregnancy on the outcomes congenital microcephaly or CNS abnormalities in a logistic regression model weighted by the propensity score.

Predictors	OR	95% CI	*p*-Value
**Exposure to ZIKV during pregnancy**	2.46	1.30–4.64	0.005
Observations: 3463 pregnant women			
R2/R2 adjusted: 0.017/0.017			
**Exposure to ZIKV during 1st trimester**	4.29	1.93–9.53	<0.001
Observations: 915 pregnant women			
R2/R2 adjusted: 0.049/0.047			
**Exposure to ZIKV during 2nd trimester**	5.29	1.08–25.95	0.040
Observations: 1793 pregnant women			
R2/R2 adjusted: 0.042/0.042			
**Exposure to ZIKV during 3rd trimester** Observations: 915 pregnant women R2/R2 adjusted: 0.049/0.047	0.68	0.21–2.14	0.506

**Table 5 viruses-17-00208-t005:** Characteristics of cases of congenital microcephaly or CNS abnormalities from mothers ZIKV+ and ZIKV−.

	ZIKV+ *n* (%)	ZIKV− *n* (%)	*p*-Value ^1^
**Total**	49	44	
**Maternal age** [median (IQR)]	25 (21–29)	23 (20–31)	0.600
**Place of residence**			
Not urban	40 (81.6)	36 (81.8)	0.982
Urban	9 (18.4)	8 (18.2)	
**Maternal ethnicity/race**			
White	18 (40)	17 (39.5)	
Others	27 (60)	25 (60.5)	0.014
Missing	1	4	
**Fetus/newborn sex**			
Female	22 (47.8)	27 (67.5)	0.066
Male	24 (52.2)	13 (32.5)	
Missing	3	4	
**Newborn birth length**			
cm [median (IQR)]	45 (43.2–47.8)	47 (44–48)	0.505
**Newborn birth weight**			
g [median (IQR)]	2640 (2402.5–2902.5)	2637.5 (2120–2957.5)	0.996
**Newborn head circumference**			
cm [median (IQR)]	30 (28–31)	29.2 (28–30,1)	0.460
**Prematurity**			
No	36 (83.7)	37 (84.1)	0.164
Yes	3 (7)	4 (9.1)	
Not applicable	4 (9.3)	3 (6.8)	
**Twinning**			
No	49 (100)	44 (100)	1
Yes	0	0	

Percentages for all categories were calculated with the exclusion of those with missing data from the denominator. The “missing” category was not included as a category when the *p*-value was estimated. IQR: interquartile range. ^1^: Chi-square test.

**Table 6 viruses-17-00208-t006:** Maternal symptoms in cases of congenital microcephaly or CNS abnormalities from mothers ZIKV+ and ZIKV−.

	ZIKV+ *n* (%)	ZIKV− *n* (%)	*p*-Value ^1^
**Total**	49	44	
**Fever**			
Yes	6 (12.2)	12 (27.3)	0.067
No	43 (87.8)	32 (72.7)	
**Rash**			
Yes	46 (93.9)	35 (79.5)	0.04
No	3 (6.1)	9 (20.5)	
**Arthralgia**			
Yes	16 (32.7)	16 (36.4)	0.707
No	33 (67.3)	28 (63.6)	
**Headache**			
Yes	14 (28.6)	15 (34.1)	0.566
No	35 (71.4)	29 (65.9)	
**Conjunctivitis**			
Yes	3 (6.1)	1 (2.3)	0.619
No	46 (93.9)	43 (97.7)	
**Coryza**			
Yes	1 (2)	4 (9.1)	0.186
No	48 (98)	40 (90.9)	
**Diarrhea**			
Yes	4 (8.2)	2 (4.5)	0.68
No	45 (91.8)	42 (95.5)	
**Retro-ocular pain**			
Yes	7 (14.3)	5 (11.4)	0.675
No	42 (85.7)	39 (88.6)	
**Edema**			
Yes	2 (4.1)	5 (11.4)	0.249
No	47 (95.9)	39 (88.6)	
**Myalgia**			
Yes	14 (28.6)	10 (22.7)	0.52
No	35 (71.4)	34 (77.3)	
**Lymphadenomegaly**			
Yes	2 (4.1)	0	0.496
No	47 (95.9)	44 (100)	
**Pruritus**			
Yes	33 (67.3)	20 (45.5)	0.033
No	16 (32.7)	24 (54.5)	
**Cough**			
Yes	0	1 (2.3)	0.473
No	49 (100)	43 (97.7)	
**Fetal loss**			
Yes	4 (9.1)	5 (10.2)	1 *
No	40 (90.9)	44 (89.8)	

Percentages for all categories were calculated with the exclusion of those with missing data from the denominator. The “missing” category was not included as a category when the *p*-value was estimated. ^1^: Chi-square test, except for those labeled with an asterisk (*), which used Fisher’s exact test.

**Table 7 viruses-17-00208-t007:** CNS abnormalities, physical examination, and ophthalmologic exam findings in ZIKV+ and ZIKV− cases.

	ZIKV+ *n* (%)	ZIKV− *n* (%)	*p*-Value ^1^
**Total**	49	44	
**CNS abnormalities:**			
Intracranial calcifications	26 (74.3)	15 (75)	1
Ventriculomegaly	23 (65.7)	19 (95)	0.033
Posterior fossa malformations	6 (17.1)	5 (25)	0.723
Reduced brain volume	6 (17.1)	4 (20)	1
Corpus callosum malformations	6 (17.1)	3 (15)	1
Cortex dysplasia	6 (17.1)	1 (5)	0.379
Lissencephaly	4 (11.4)	3 (15)	1
Pachygyria	3 (8.6)	1 (5)	1
Hydrops fetalis	1 (2.9)	0	1
Cystic hygroma + encephalocele	0	1 (5)	0.775
Semilobar holoprosencephaly	0	1 (5)	0.775
**Physical examination findings:**			
Arthrogryposis	4 (8.2)	0	0.154
Congenital foot deformities	1 (2)	1 (2.3)	1
Esophageal atresia	1 (2)	0	1
Cleft lip and palate	0	1 (2.3)	0.957
Myelomeningocele	0	1 (2.3)	0.957
**Ophthalmologic examination:**			
Optic nerve hypoplasia	4 (44.4)	3 (30)	0.514
Incomplete vascularization	1 (11.1)	0	0.279
Pigmentary abnormalities	2 (22.2)	0	0.115
Retinal coloboma	0	2 (20)	0.156
Chorioretinal atrophy	0	1 (10)	0.329
Chorioretinitis	0	1 (10)	0.329
Microphthalmia	0	1 (10)	0.329

Percentages for all categories were calculated with the exclusion of those with missing data from the denominator. The “missing” category was not included as a category when the *p*-value was estimated. ^1^: Chi-square test.

## Data Availability

In our study, we used five databases: GAL, FORMSUS, RESP, SINASC, and SIM, all provided by the Rio de Janeiro State Health Department. Data may be made available upon reasonable request by the corresponding author.

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
