# Peer review of "Neonatal Microcephaly and Central Nervous System Abnormalities During the Zika Outbreak in Rio de Janeiro"

_viruses, 2025, doi:10.3390/v17020208_

Round 1
Reviewer 1 Report
Comments and Suggestions for Authors
This study makes an important contribution to understanding the impact of ZIKV infection during pregnancy on fetal and newborn health, focusing on microcephaly and central nervous system (CNS) abnormalities. Its significance lies in the integration of five public health databases (GAL, FORMSUS, RESP, SINASC, and SIM), enabling a comprehensive examination of maternal and neonatal outcomes during the ZIKV outbreak in Brazil. The use of probabilistic linkage among these datasets represents a methodological strength, allowing for a more detailed and cohesive analysis of outcomes.
The study findings strongly support the conclusion that ZIKV infection during pregnancy—particularly in the first and second trimesters—significantly increases the risk of microcephaly and CNS abnormalities in newborns, aligning with prior research on the virus's impact on fetal development.
While the study is robust, it openly acknowledges several inherent limitations. The lack of serological testing for other congenital infections, such as cytomegalovirus (CMV) and toxoplasmosis, in some cases reflects practical challenges in large-scale data collection. Another key point is that most cases of ZIKV infection analyzed were symptomatic, which may limit the generalizability of findings to asymptomatic cases. Lastly, the absence of complementary examinations in all identified cases could reduce the ability to confirm other potential causes of congenital abnormalities.
Despite these limitations, which were addressed by the authors, the study's design, methodological rigor, and large dataset ensure its relevance and reliability in advancing our understanding of ZIKV's impact on maternal and neonatal health.
Comments on the Quality of English LanguageThe English quality is good, but there is room for improvement in:
-
Grammar and punctuation (adding articles and addressing typos).
-
Sentence structure (ensuring smooth transitions and consistency in tense).
-
Word choice and phrasing (avoiding redundancy and improving readability).
Areas for Improvement
Grammatical Issues:
Example: "Eligibility for confirmed ZIKV infection cohort required at least one positive RT-PCR assay..."
Suggestion: Add "the" before "confirmed ZIKV infection cohort" to maintain grammatical correctness.
Example: "All the databases were provided by the Healthy Secretary of Rio de Janeiro, and the study protocol received ethical approval..."
Suggestion: Change "Healthy Secretary" to "Health Secretary" (presumed typo).
Phrasing and Word Choice:
Example: "Finally, a probabilistic linkage was performed among the five databases."
Suggestion: "Probabilistic linkage" could be rephrased for better readability if a broader audience is intended.
Example: "For the logistic model, the exposure effect is an adjusted odds ratio."
Suggestion: Clarify the sentence to ensure smooth readability.
Reviewer 2 Report
Comments and Suggestions for Authors
The study is aimed to describe in details the ZIKV outbreak in the State of Rio de Janeiro and its impacts on pregnant women and their babies.
The authors identified a large number (2 269) pregnant women with at least one positive result for ZIKV infection followed by 49 cases of congenital microcephaly or CNS abnormalities were identified (2.16% with 0.78% in the control group).
The estimated odds ratio of congenital microcephaly or CNS abnormalities caused by exposure to ZIKV during pregnancy was 2.46.
Social and demographic data for the studied population are presented. Detailed medical data regarding pregnancy, time of diagnosis antenatal information etc. summarised in the tables. Spectrum of CNS abnormalities is also presented with the most common findings among ZIKV+ cases encountered intracranial calcifications, ventriculomegaly, posterior fossa malformations, reduced brain volume, corpus callosum malformations, cortex dysplasia, lissencephaly, and pachygyria. The other congenital abnormalities (musculoskeletal, ocular) are also presented and discussed.
The paper reflects the real clinical and epidemiological practice in the region, the research question is clearly answered, the paper is well structured narrated, statistical analysis is sufficient.
As the authors conclude, since ZIKV infections still endemic in some countries globally, including Brazil, reinforcing the preventive measures and continuous monitoring of potential fetal anomalies. The paper is recommended for the publication in the present version.
Reviewer 3 Report
Comments and Suggestions for Authors
Dear authors,
The manuscript "Microcephaly and central nervous system abnormalities during the Zika Outbreak in Rio de Janeiro” provides valuable information about the congenital abnormalities of nervous system during the Zika virus outbreak. This is a retrospective study that analyzed 7,870 cases of pregnant women in the state of Rio de Janeiro. The work is clear and well written. The authors have highlighted both the strengths and weaknesses of the publication. Due to the value of the publication and the importance of the topic, I recommend the manuscript for publication.
Below are some minor comments:
Lines 25: unnecessary % sign
In Table 3, in the 16th row, the line from Table 1 is pasted: "Prevalence 1.5% 0.36%".
Line 123: had high education => had high school education
Line 123: “They also had natural labor…” According to Table 3, cesarean section deliveries were more frequent in both groups. Please check the correctness of the data in the table and text.
Thank you.
Reviewer 4 Report
Comments and Suggestions for Authors
In this article, Marlos Melo Martins and Colleagues have conducted a retrospective cohort study on the effect of Zika (ZIKV) infection on central nervous system (CNS) abnormalities of fetuses in pregnant women, a condition described as congenital Zika Syndrome (CZS). The data were collected between February 2015 and December 2018 and containing information about pregnant women and their neonates during the ZIKV outbreak in Rio de Janeiro, Brazil. This study included 7,870 pregnant women (2,269 positive to RT-qPCR for ZIKV and 5,601 negative to RT-qPCR for ZIKV). Among the ZIKV-positive pregnant women, 49 cases of congenital microcephaly or CNS abnormalities were identified (2.16% or 193.9 cases in 10 000 live births), whereas 44 cases were identified among ZIKV-negative women (0.78% or 71.4 cases in 10 000 live births). The odds ratio of congenital microcephaly or CNS abnormalities caused by exposure to ZIKV during pregnancy was 2.46 (95% CI 1.30-4.64) and when the pregnant women were stratified according to the gestation's trimester, the following odd ratio were observed: 4.29 (95% CI 1.93-9.53) in the first trimester, 5.29 (95% CI 1.08-25.95) in the second trimester and 0.68 (95% CI 0.21-2.14) in the third trimester. The most common findings among ZIKV-positive cases were intracranial calcifications, ventriculomegaly, posterior fossa malformations, reduced brain volume, corpus callosum malformations, cortex dysplasia, lissencephaly, and pachygyria. Ophthalmologic exams were performed in nine ZIKV-positive newborns (55.5% abnormal) and ten ZIKV-negative newborns (30% abnormal). Brainstem auditory evoked potentials (BAEPs) were reported in nine ZIKV-positive cases (33.3% abnormal) and in six ZIKV-negative cases (16.6% abnormal). The Authors concluded that monitoring neurological, motor, auditory, visual, and orthopedic disorders must be conducted in all children intrauterine exposed to ZIKV, especially to those exposed during the first and the second trimesters of pregnancy. The study is well-structured, and the analysis performed properly. However, I would like to point out some minor revisions to address:
1. In the abstract, I would check the last sentence. I would also provide a final sentence with the main conclusion obtained in this study.
2. I would also revise the title as: Neonatal microcephaly and central nervous system abnormalities during the Zika Outbreak in Rio de Janeiro.
3. The Authors have improperly used the word concepts all over the manuscript. I would suggest replacing this word with more appropriate words like fetuses, newborns or neonates, that must be chosen accordingly. In English, concepts do not mean a recently born children or an offspring of a human in the stages of prenatal development that follow the embryo stage. Concepts means an abstract idea or a general notion.
4. In the Table 3, to assess the weigh of ethnicity/race, place of residence and twin pregnancy on ZIKV infection, a chi-square test was performed, whereas for the same categories in Table 5, a Fisher’s exact test or a Mann-Whitney U test was performed. The Authors should clarify which method has been used for the selection of the appropriate statistical test.
5. I would suggest a revision of the English language.
